# Genetic Diversity and Classification of *Colletotrichum sublineola* Pathotypes Using a Standard Set of Sorghum Differentials

**DOI:** 10.3390/jof10010003

**Published:** 2023-12-20

**Authors:** Louis K. Prom, Ezekiel Jin Sung Ahn, Ramasamy Perumal, Hugo E. Cuevas, William L. Rooney, Thomas S. Isakeit, Clint W. Magill

**Affiliations:** 1Crop Germplasm Research Unit, USDA-ARS, 2881 F & B Road, College Station, TX 77845, USA; 2Plant Science Research Unit, USDA-ARS, St. Paul, MN 55108, USA; ahn00079@umn.edu; 3Department of Agronomy, Agricultural Research Center, Kansas State University, Hays, KS 67601, USA; perumal@ksu.edu; 4Tropical Agriculture Research Station, USDA-ARS, 2200 Pedro Albizu Campos Avenue, Mayaguez, PR 00680, USA; hugo.cuevas@usda.gov; 5Department of Soil and Crop Sciences, Texas A&M University, College Station, TX 77843, USA; william.rooney@ag.tamu.edu; 6Department of Plant Pathology and Microbiology, Texas A&M University, College Station, TX 77843, USA; thomas.isakeit@ag.tamu.edu (T.S.I.); c-magill@tamu.edu (C.W.M.)

**Keywords:** sorghum anthracnose, genetic variability, pathogenicity, pathotype, *Colletotrichum sublineola*

## Abstract

Anthracnose, incited by *Colletotrichum sublineola,* is the most destructive foliar disease of sorghum and, under severe conditions, yield losses can exceed 80% on susceptible cultivars. The hyper-variable nature of the pathogen makes its management challenging despite the occurrence of several resistant sources. In this study, the genetic variability and pathogenicity of 140 isolates of *C. sublineola,* which were sequenced using restriction site-associated sequencing (RAD-Seq), resulted in 1244 quality SNPs. The genetic relationship based on the SNP data showed low to high genetic diversity based on isolates’ origin. Isolates from Georgia and North Carolina were grouped into multiple clusters with some level of genetic relationships to each other. Even though some isolates from Texas formed a cluster, others clustered with isolates from Puerto Rico. The isolates from Puerto Rico showed scattered distribution, indicating the diverse nature of these isolates. A population structure and cluster analysis revealed that the genetic variation was stratified into eight populations and one admixture group. The virulence pattern of 30 sequenced isolates on 18 sorghum differential lines revealed 27 new pathotypes. SC748-5, SC112-14, and Brandes were resistant to all the tested isolates, while BTx623 was susceptible to all. Line TAM428 was susceptible to all the pathotypes, except for pathotype 26. Future use of the 18 differentials employed in this study, which contains cultivars/lines which have been used in the Americas, Asia, and Africa, could allow for better characterization of *C. sublineola* pathotypes at a global level, thus accelerating the development of sorghum lines with stable resistance to the anthracnose pathogen.

## 1. Introduction

Sorghum (*Sorghum bicolor* (L.) Moench) is a versatile crop, and its adaptability in marginal agro-ecological zones makes it indispensable for people and animals living in dry tropical regions [1,2,3,4,5]. Among cereals, sorghum acreage and production rank behind those of maize, rice, wheat, and barley, and its uses include human consumption, notably, in the health food industry, animal feed, and biofuel [1,4,6,7]. The adaptability of sorghum to a wide range of environments exposes the crop to diverse abiotic and biotic stresses [1,8]. Abiotic stresses such as drought and high temperatures are critical factors that limit sorghum yield performance in drier tropical zones [8]. Among the biotic stresses, *Colletotrichum sublineola* (formerly *C*. *graminicola* P. Henn in Kabàt and Bubk), the causal agent of sorghum anthracnose, is the most important foliar disease in sorghum, because the pathogen infects all the above-ground plant parts such as panicle, stalk, and grain [9,10]. The foliar phase of the disease is the most damaging, resulting in yield losses of up to 86% [11]. Infection of the stalk results in stalk rot, which may lead to lodging and lower harvestable biomass [12], while panicle infection can result in grain losses of up to 50% [9]. Management options for sorghum anthracnose include crop rotation, application of fungicides, and the use of resistant cultivars [9,10,13,14]. The use of resistant cultivars is the most effective strategy for controlling anthracnose because it lowers production costs and is environmentally friendly [4,9,10,13,15]. However, the hypervariable nature of the *C. sublineola* pathogen requires selection for resistance based on the specific pathotypes in a target environment [9,15,16,17,18]. In India, Pande et al. [16] tested the pathogenicity of nine anthracnose isolates on thirty sorghum genotypes and reported nine distinct pathotypes. Moore et al. [17] evaluated ninety-eight isolates from Arkansas, USA, on eight sorghum lines and documented thirteen different pathotypes. Although there a morphological variations among the isolates of *C*. *sublineola,* these characteristics do not elucidate the pathogenic differences in the host–pathogen interaction [9,14,16,18]. Recently, Koima et al. [14] evaluated seven *C. sublineola* isolates with different morphological and cultural characteristics using a detached leaf assay and found no differences in pathogenicity on the sorghum cultivar Kateng’u.

Due to environmental influences on the stability of morphological traits, differentiation among *Colletotrichum* isolates based on conidial morphology, such as colony color, size, shape, or host origin, is insufficient to assess genetic diversity. Hence, molecular markers have been used to examine diversity in the pathogen [18]. Over the years, the genetic diversity of *C*. *sublineola* isolates has been reported by several researchers using polymorphic DNA markers such as restriction fragment length polymorphism (RFLP), random amplified polymorphic difference (RAPD), and amplified fragment length polymorphism (AFLP) [14,18,19,20,21,22,23,24]. Prom et al. [18] reported high variability among 232 *C*. *sublineola* isolates collected from the U.S. and Puerto Rico using AFLP analysis, while Chala et al. [25] noted the existence of diversity among 22 isolates collected from a single sorghum field in Ethiopia. A total of 384 isolates collected from sorghum and Johnsongrass (*Sorghum halepense* (L.) Pers.) from the U.S., Burkina Faso, Zambia, South Africa, Sudan, Brazil, and Puerto Rico were characterized using RFLP and RAPD fingerprinting by Xavier et al. [24]. While many studies have been completed, none have been completed with single nucleotide polymorphism (SNP) markers from next-generation sequencing (NGS) facilities.

To effectively deploy resistance sources, knowledge of the pathotypes of *C. sublineola* in a region is essential. The existence of pathotypes and environmental factors can partially elucidate the differential reactions of sorghum lines that are deployed in different production regions or evaluated in different fields [8,18]. Given that new virulent pathotypes of *C. sublineola* will occur, their monitoring is of paramount importance for host plant resistance in sorghum. In this current study, the aim was to determine the range of pathotype variation in *C. sublineola* using a set of sorghum genotypes known to react differently to anthracnose (Table 1) and sequence 140 *C*. *sublineola* isolates to determine genetic diversity through SNP markers.

## 2. Materials and Methods

### 2.1. Isolation and Storage of C. sublineola Isolates

Leaf samples from sorghum infected with anthracnose (as verified by the presence of acervuli) were collected from research plots and production fields between 2006 and 2016 in Georgia (Cairo, Fitzgerald, and Tifton), North Carolina (Winterville), Puerto Rico (Corozal, Isabela, and Mayaguez), and Texas (Burleson and Wharton counties) (Table 2). The samples were stored in a refrigerator until they were ready for culturing. Following the protocol established by Prom et al. [18], single spore isolates were placed on separately sterilized Whatman No. 2 filter paper (Whartman International LTD, Maidstone, England, UK) placed on a half-strength potato dextrose agar and incubated at 25 °C over a 12-h photoperiod until the paper was colonized. The colonized papers were cut, put in separate sterile vials, and stored at −20 °C at the USDA-ARS, Southern Plains Agricultural Research Center, College Station, TX, USA, for long-term storage.

### 2.2. Fungal DNA Extraction, Restriction Site-Associated Sequencing (RAD-Seq), and Phylogeny Reconstruction

DNA samples from fungal isolates were obtained using the method described by Prom et al. [18]. In brief, mycelium was rinsed 2 to 3 times with 0.1 M MgCl_2_ and dried (10–15 min) using a Savant SpeedVac DNA 110 (GMI, Ramsey, MN, USA). A MasterPure^TM^ Yeast DNA Purification kit (Biotechnologies, Thermo Fisher Scientific, Austin, TX, USA) was used to extract DNA from 138 isolates of *C. sublineola* as well as JG1 and JG2 from Johnsongrass in Corpus Christi, Texas. After a quality and quantity check, all the DNA samples were sent to the Genomics and Bioinformatics Service component of Texas A&M AgriLife Research for restriction site-associated sequencing (RAD-Seq). Each sample was bar-coded and sequenced from each end of the restriction fragments using ILLUMINA technology, San Diego, CA, USA). The number of approximately 120 base pair reads per sample ranged from 780,000 to nearly 7 million (12X coverage). These were provided pre-screened to assure high quality, with the primer adaptor and barcode sequences already stripped. Tools in the CLC Genomics Workbench (v8) were used to align the sequences from each read to the sequenced contigs from a rough draft of the *C. sublineola* genome entered into GenBank by Baroncelli et al. [34]. A subset of 1244 SNPs with missing data < 10% and minor allele frequency > 0.05 were retained for a population and phylogenetic analysis.

#### Population Structure and Cluster Analysis

The population structure of the *C. sublineola* isolates was determined using the model-based clustering method implemented in STRUCTURE 2.1 [35]. Ten independent runs using an admixture model with correlated frequencies, 25,000 burn-in periods, and 125,000 Monte Carlo Markov Chain (MCMC) were completed for each k value set from 1 to 13. The ad hoc statistic Δ*k* based on the rate of change in the log probability of data [36] and the observed convergency in the mean of the log probability of the data between successive *k* values, both as implemented by the Structure Harvester software (https://taylor0.biology.ucla.edu/structureHarvester/) [37], were used to fraction the genetic variance into populations. The ten independent runs of the selected *k* values were matched in CLUMPP [38] to obtain the ancestry membership coefficient of each isolate. The isolates with an ancestry coefficient > 0.75 were assigned to their corresponding population. A principal component analysis (PCA) was conducted in Tassel 5.0 (TASSEL-GBS), which, according to Glaubitz et al. [39], allows for high-throughput genotyping of large numbers of individuals at a considerable number of SNP markers.

The identical-by-state (IBS) genetic distances among the 140 *C. sublineola* isolates were calculated in Tassel 5.0 and subjected to a clustering analysis using neighbor-joining. The phylogenetic tree was visualized using Interactive Tree of Life [40].

### 2.3. Host Differentials and Isolates Used for Pathotype Determination

A total of 18 sorghum differentials—RTx2536, SC748-5, BTx398, TAM428, RTx430, Brandes, SC112-14, Theis, BTx378, SC326-6, SC283, BTx623, SC328C, SC414-12E, PI570841, PI570726, PI569979, and IS18760—were used in this study and are described in detail by Prom et al. [18].

Based on the phylogeny analysis, a total of 30 diverse isolates (Georgia: FSP213, FSP276, FSP277, FSP278, FSP279, FSP280, FSP281, FSP284, FSP289, and FSP299; North Carolina: FS265 and FSP267; Puerto Rico: FSP70, FSP71, FSP76, FSP92, FSP198, FSP199, FSP200, FSP201, FSP208, FSP244, FSP245, FSP248, FSP250, and FSP252; and Texas: FSP36, FSP53, FSP182, and FSP237) were arbitrarily selected from the phylogenetic analysis and evaluated in the greenhouse for virulence against sorghum differential lines.

### 2.4. Greenhouse Experiment

The protocol for the greenhouse experiment, inoculum preparation, inoculation, and disease assessment were described by Prom et al. [18,41]. Briefly, the greenhouse experimental design was a split-plot with 30 *C. sublineola* isolates as the main plot and 18 sorghum differentials as the sub-plot. Seeds from each differential were planted at a rate of eight seeds per tall tree pot (4″ × 14″) (Hummert International) with metro mix 200 (BWI) containing potting soil mixed with osmocote classic fertilizer 17-7-12 (O.M. Scott & Sons Company, Marysville, OH, USA). Each differential line (RTx2536, SC748-5, BTx398, TAM428, RTx430, Brandes, SC112-14, Theis, BTx378, SC326-6, SC283, BTx623, SC328C, SC414-12E, PI570841, PI570726, PI569979, IS18760) was replicated three times. To accommodate the space in the greenhouse, four tall tree pots were placed in 3-gallon poly-trainer cans (10″ × 91/2″ × 85/8″) (Hummert International). Germinated plants at the three-leaf stage were thinned to four plants per pot. A total of 200 mL of Peters Excel 15-5-15 (O.M. Scott & Sons Company, Marysville, OH, USA) multi-purpose fertilizer was applied to each tall pot on a bi-weekly basis pre-inoculation. At the eight-leaf stage, eight *C. sublineola*-colonized seeds were placed in each plant whorl and, later in the evening, the plants were inoculated with 1 × 10^6^ conidia/mL suspension until run-off with their respective isolate. To create a favorable condition for disease development, the plants were misted for 30 s at 45 min intervals for 8 hrd^−1^ for one month. The experiments were repeated twice.

### 2.5. Disease Assessment and Data Analysis

The plants were assessed for anthracnose infection twice, 30 days post-inoculation and a week later, using the Prom et al.’s [18,41] disease rating scale 1–5, as follows: 1 = no symptoms or chlorotic flecks on leaves; 2 = hypersensitive reaction (reddening or red spots) on inoculated leaves but no acervuli formation; 3 = lesions on inoculated and bottom leaves with acervuli in the center; 4 = necrotic lesions with acervuli observed on inoculated and bottom leaves with infection spreading to middle leaves and not yet on the flag leaves; and 5 = most leaves dead due to infection with infection on the flag leaf containing abundant acervuli. The symptom types were then categorized into the following two reaction classes: resistant = rating 1 or 2; and susceptible = rating 3, 4, or 5. The data on the anthracnose rating were analyzed using the command PROC ANOVA (SAS Institute, SAS version 9.4, Cary, NC, USA).

## 3. Results

The SNP data from 140 *C. sublineola* isolates were only partially grouped by origin (Figure 1). For example, isolates from Burleson County, Texas, were interspersed among the isolates from Puerto Rico, while the isolates from Wharton County, Texas, were grouped in with the isolates from Georgia and North Carolina (Figure 1). The isolates from Puerto Rico showed the most widespread genetic diversities, and the isolates from Johnsongrass, JG-1 and JG-2, grouped together with a 100% bootstrap consensus value and were close to multiple isolates from Georgia.

### 3.1. Genetic Diversity of C. sublineola

The genetic diversity of *C. sublineola* varied across locations. The population structure analysis based on Δ*k* stratified the genetic diversity into two large populations (Figure 1). We observed that the isolates from Puerto Rico and some from Texas were genetically related and clustered into one population, while the other isolates from Georgia, North Carolina, and Texas constituted another population. To obtain additional insight into the genetic variation of *C. sublineola*, the genetic variation was also stratified into eight populations (105 isolates) and one admixture group (35 isolates) based on the mean variation of the log probability of the data (Figure 2). This analysis showed that isolates from Georgia, North Carolina, and Texas could be separated into six groups, the two isolates from Johnsongrass in one group, and the isolates from Puerto Rico and Texas in one large group. This population structure was also observed in the principal component analysis, in which isolates from Johnsongrass were located at the center, surrounded by a group of isolates from Georgia and North Carolina. Remarkably, the isolates from Tifton and Cairo, GA, constitute two distinct groups, suggesting that both exhibit unique genetic variation.

The phylogenetic analysis was consistent with the population structure analysis (Figure 2). The isolates from Puerto Rico and Texas were clustered into one main clade, separated from other clades by admixtures isolates. The isolates from Tifton, GA, were the most genetically related to the isolates from Johnsongrass. We observed that the isolates from Cairo, GA, are distributed into three clades, of which one includes three isolates from North Carolina. Certainly, the genetic variation of *C. sublineola* isolates is associated with the agri-environmental niches of each location.

### 3.2. Virulence of C. sublineola Isolates

The main effects of the isolates (Georgia: FSP213, FSP276, FSP277, FSP278, FSP279, FSP280, FSP281, FSP284, FSP289, and FSP299; North Carolina: FS265 and FSP267; Puerto Rico: FSP70, FSP71, FSP76, FSP92, FSP198, FSP199, FSP200, FSP201, FSP208, FSP244, FSP245, FSP248, FSP250, and FSP252; and Texas: FSP36, FSP53, FSP182, and FSP237) and the differential interaction between the sorghum differential lines (RTx2536, SC748-5, BTx398, TAM428, RTx430, Brandes, SC112-14, Theis, BTx378, SC326-6, SC283, BTx623, SC328C, SC414-12E, PI570841, PI570726, PI569979, IS18760) (*p* < 0.0001) and the isolates were highly significant, indicating that the lines responded differently when challenged with the individual isolates (Table 3).

The virulence pattern of the 30 isolates tested on the 18 sorghum differential lines revealed the existence of 27 pathotypes (Table 4). The reaction of the differential lines was the same when inoculated with FSP76 and FSP92 from Puerto Rico and FSP265 from North Carolina and, thus, was characterized as pathotype 5. Similarly, FSP245 and FSP250 isolates from Puerto Rico were designated as pathotype 15 based on the reaction of the differential lines. Pathotype 2 (FSP53 from Texas) was differentiated from pathotype 11 (FSP208 from Puerto Rico) as the former isolate-infected Theis (Table 4). Pathotypes 26 and 27 (isolates FSP289 and FSP299 from Georgia) had similar infection patterns on the sorghum differentials, except that TAM428 was resistant to isolate FSP289 (pathotype 26). Pathotype 1 (FSP36 from Texas) and pathotype 10 (FSP201 from Puerto Rico) were different based on the ability of pathotype 10 to infect SC328C and RTx2536. Pathotypes 4 and 8 (FSP71 and FSP199 from Puerto Rico) were differentiated based on the susceptibility of IS18760 and RTx2536 to isolate FSP199. Of the 27 pathotypes characterized in this study, pathotypes 3, 9, and 16 (FSP70, FSP200, and FSP248 from Puerto Rico), pathotype 13 (FSP237 from Texas), and pathotypes 20 and 23 (FSP277 and FSP280 from Georgia) were the most aggressive, infecting nine or more out of the eighteen host differentials in this study (Table 4).

The sorghum differential lines SC748-5, SC112-14, and Brandes were resistant to all the isolates evaluated, while BTx623 was susceptible to the same isolates. The host differential line QL3 (India) was resistant to all the isolates except for FSP70 from Puerto Rico and FSP237 from Texas (designated as pathotypes 3 and 13, respectively). The host differential PI570841 was susceptible to all the isolates except for FSP53 from Texas (designated as pathotype 2) and FSP208 from Puerto Rico (designated as pathotype 11).

## 4. Discussion

The expected increase in global population coupled with climate change and environmental degradation requires an increase in cereal production, including sorghum for food, feed, and other uses [42,43,44,45]. Sorghum is a drought-tolerant and low-input crop that is part of the daily food supply of millions of people, especially in dry tropical regions [1,5,44].

This crop is hampered by several fungal pathogens, including *C*. *sublineola*, the causal agent of sorghum anthracnose, which is the most important foliar disease of this crop worldwide [9,10,45]. Although many anthracnose-resistant sources have been identified, the management of this disease can be problematic due to the prevailing climatic patterns in the different agro-ecological zones where sorghum is grown and the hyper-variability of the pathogen as characterized by the use of sorghum differentials and molecular tools [9,11,13,14,18,20,22,24,29,46].

Using sequence-based SNP data, we studied the genetic diversity of the anthracnose isolates collected from different regions. Based on the genetic relationship, isolates from Puerto Rico and Texas were genetically related, while isolates from Georgia and North Carolina constituted another main population. This could be due to the historic exchange of sorghum germplasm between Puerto Rico and Texas, while Georgia and North Carolina are geographically in close proximity. Likewise, the isolates from Georgia showed similar virulence patterns to each other; as an example, even though not identical, the isolates from Georgia that formed a group, FSP279, FSP280, and FSP281, showed similar virulence patterns. Similarly, Georgia isolates FSP276, FSP277, and FSP278 showed high genetic similarities and pathotypes. In contrast, FSP76, FSP92 (Puerto Rico), and FSP265 (North Carolina) were all grouped in pathotype 5, but FSP265 was not shown to be genetically close to the other two isolates.

The isolates from Puerto Rico and Texas were highly diverse, while the isolates from Georgia and North Carolina were less so. The diversity of the pathogen in Puerto Rico, a tropical region where conditions are more favorable for anthracnose development, coupled with the fact that the isolates were collected from test plots planted with diverse sorghum germplasm could partly explain the high variability within the isolates. In contrast, the proximity of Georgia and North Carolina with similar climatic classification as well as similar sorghum hybrids in the regions may elucidate the low variability of pathogenic population. The isolates from Georgia were clustered in a tighter group with lower levels of variability. This could be attributed to the fact that the isolates were collected from the same climatic zone with the same cropping system and hybrids and possibly low prevalence and intensity of sorghum anthracnose.

In addition, other species in the genus *Colletotrichum* cause anthracnose on many economically important plants, including chili (*Capsicum* spp.), mango (*Mangifera indica*), orange (*Citrus* spp.), and strawberry (*Fragaria ananassa*) [47]. Within several *Colletotrichum* spp., pathogenic variation based on pathogenicity on sets of host differentials has been documented [48,49]. In a previous study by Prom et al. [18], 17 pathotypes were established from 20 diverse isolates using 18 sorghum differentials, including nine lines previously used by Casela and Ferreira [26]. In the current study, 27 new pathotypes were distinguished using 30 sequenced diverse isolates collected from Georgia, North Carolina, Puerto Rico, and Texas and evaluated with the same 18 sorghum differentials used earlier by Prom et al. [18]. Similar host–*Colletotrichum* spp. studies in Brazil, resulted in five pathotypes of *C*. *graminicola* when the virulence pattern of 190 isolates on 15 maize differentials was observed [49]. Montri et al. [48] documented three pathotypes of *C. capsici* out of eleven isolates using nine chili differentials. In this study, Brandes, SC748-5, and SC112-14 were resistant to all the *C*. *sublineola* isolates tested. However, Tsedaley et al. [50] observed that SC748-5 was susceptible to one of the five isolates from Ethiopia tested in the greenhouse.

Although only 30 isolates collected from four climatic zones were tested, many pathotypes were identified, confirming the hyper-variable nature of *C*. *sublineola.* Yet, no association was either noted or inference made between specific climatic zones and pathotype. In the present study, the isolates within each population group revealed high levels of variability for the genes affecting pathogenicity on the sorghum differentials. Similarly, high levels of variability have also been noted among 232 *C*. *sublineola* placed in four clusters based on AFLP analysis [18]. Using RAPD and RFLP-PCR markers to evaluate the genetic diversity among 37 sorghum anthracnose isolates collected from Brazil, Valèrio et al. [22] observed no association between virulence patters and molecular profiles. Also, a RAPD analysis of 19 *C*. *lupini* isolates detected high intraspecific genetic diversity with marked differences in pathogenicity on susceptible cultivar ‘kiev mutant’ [51]. However, the clustering of *Xanthomonas translucens* pv. *undulosa* and *X*. *translucens* pv. *translucens* based on multilocus sequencing typing and multilocus sequencing analysis showed correlations among the strains and levels of virulence on inoculated wheat and barley [52]. This study suggests that molecular tools to determine genetic diversity could be used to predict relative virulence on some host pathosystems. In the sorghum anthracnose pathosystem, Chala et al. [23] suggested that certain factors such as geographic separation, diverse sorghum lines planted, and the different agro-ecological zones where the crops are planted may contribute to the evolution and diversity of *C*. *sublineola*. However, other mechanisms that contribute to fungal population diversity include mutation, sexual reproduction, gene gain or loss, gene family expansion and contraction, transposable elements, loss of heterozygosity, copy variation, etc. [53,54]. Some, if not all, of these mechanisms may also be operating in the *C*. *sublineola* pathogen population.

Further, due to the existence of a large number of *C*. *sublineola* pathotypes, continuous evaluation of sorghum germplasm and robust monitoring of any changes in the pathogenic population, coupled with the use of a standard set of differentials to compare pathotypes, would help researchers identify stable sources of anthracnose resistance. Additionally, crosses among sorghum differentials and the study of their inheritance may lead to our understanding of whether the gene-for-gene concept operates in this host–pathogen interaction.

## Figures and Tables

**Figure 1 jof-10-00003-f001:**
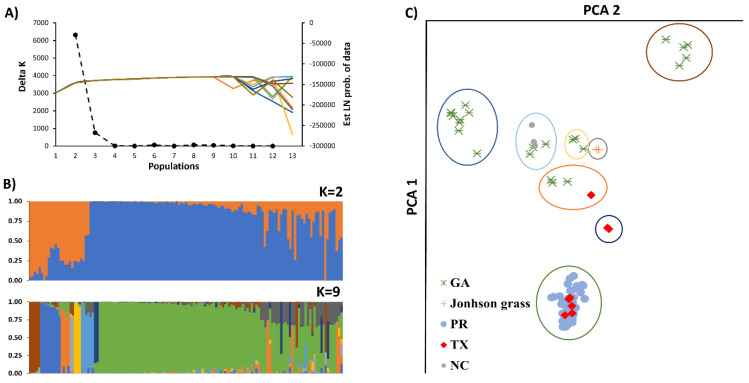
Population structure analysis of one hundred and thirty-eight *C. sublineola* isolates collected in sorghum fields in Georgia (GA), Texas (TX), North Carolina (NC), and Puerto Rico (PR) and two isolates collected from Johnsongrass in Corpus Christi, Texas, using 1244 SNPs. (**A**) Estimation of the number of populations in the 140 *C. sublineola* isolates based on the analysis in STRUCTURE, with Δ*k* values (Axis 1; black dashed line) and the estimate LN probability of data (axis 2; different color line per each STRUCTURE run) using 10 runs for each K values from 1 to 13; (**B**) Hierarchical organization of genetic relatedness of 140 *C. sublineola* isolates for K values of 2 and 9. (**C**) Principal component analysis of the 105 *C. sublineola* isolates present in the eight populations found in the STRUCTURE analysis.

**Figure 2 jof-10-00003-f002:**
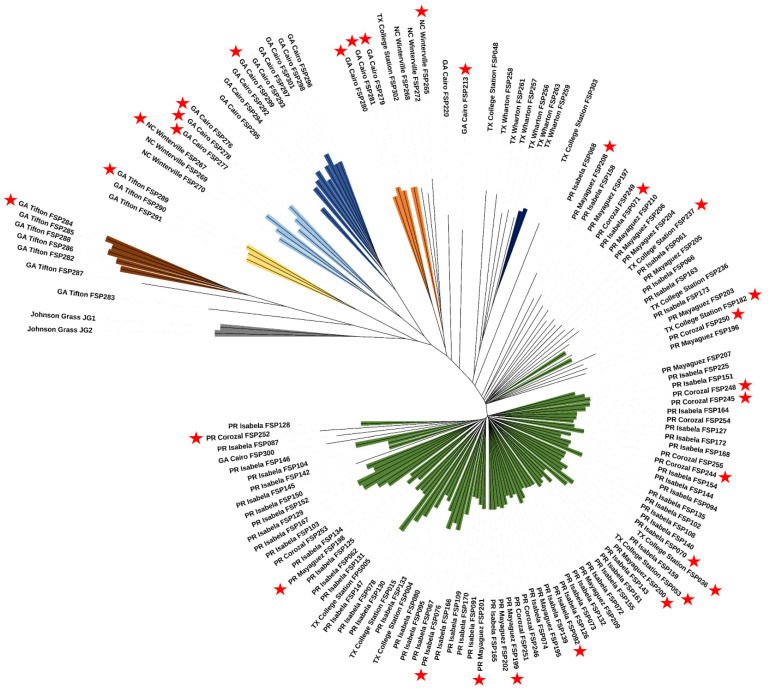
Unrooted neighbor-joining tree for 140 *C. sublineola* isolates collected in Georgia, Texas, North Carolina, and Puerto Rico. Colored branches represent isolates belonging to the eight populations found in the STRUCTURE analysis, while admixture isolates are not colored. The red stars represent isolates used for a virulence analysis against 18 sorghum differential lines.

**Table 1 jof-10-00003-t001:** Sorghum differentials used in prior studies, number of pathotypes identified, and references ^a^.

Sorghum Differentials Tested	Number of Pathotypes Identified	Reference
BTx378, SC326-6, SC283, BTx623, Brandes, SC112-14, BTx398, RTx2536, Theis	16	[26]
IS4225, IS8361, Br64, 954206, 954130, 954062	3	[27]
RTx2536, BTx398, TAM428, RTx430, SC414-12E, BTx378, SC326-6, QL3 (IND)	8	[28]
IS643, IS854, IS914, IS1006, IS1022, IS2058, IS2596, IS3089, IS3589, IS5511, IS6958, IS7142, IS7775, IS8024, IS8283, IS9600, IS12467, IS12664C, IS17141, IS17804, IS18433, IS18442, IS18521, IS18531, IS18615, IS18680, IS18688, IS18758, IS18760, UChV-2	9	[16]
RTx2536, BTx398, BTx623, Brandes, SC112-14, Theis, BTx378, SC326-6, SC283	8 groups designated with letters (A–H) with 32 races in each group	[29]
IS8354, IS2508, IS18758, IS3738, IS6928, IS18760, IS3552, IS854, IS1006, IS18442, IS6958, IS12467, IS17141, IRAT204, ICSV247, A2267-2	6	[30]
KVS8, BES, IS3758, IS6926, IRAT204, IS6958, IS18442, A2267-2	7	[31]
RTx2536, BTx398, TAM428, SC414-12E, BTx378, SC326-6, SC328C, QL3 (IND)	13	[17]
RTx2536, SC748-5, BTx398, TAM428, RTx430, Brandes, SC112-14, Theis, BTx378, SC326-6, SC283, BTx623, SC328C, SC414-12E, PI570841, PI570726, PI569979, IS18760	17	[18]
PU932247, BTx378, PUGP24, Pop.BRP3R9, Tx614, CMSXS169 IPB8030, SC326-6, BTx623)	68	[32]
Bailey, Chinese Amber, Cowper, Dale, Yellow milo, Honey, N100, Orange, Planter, Simon, Della, and Keller	12	[24]

^a^ The sorghum differentials used in the present study were previously compiled by Prom et al. [18], RTx2536, SC748-5, Martin (BTx398), TAM428, BTx430, Brandes, SC112-14, Theis, BTx378, SC326-6, SC283, BTx623, SC328C, SC414-12E, PI570841, PI570726, PI569979, IS18760.

**Table 2 jof-10-00003-t002:** Details of *Colletotrichum sublineola* isolates evaluated in this study ^a^.

Code for the Isolate	Year of Collection	Total	Location and Coordinates	Climate
FSP182	2006	11	College Station, TX, USA 30°36′99″ N–96°18′31.20″ W	Subtropical and temperate
FSP4, 5, 15, 36, 48, 53	2011			
FSP236, 237	2012			
FSP302, 303	2016			
FSP158, 159, 161, 163–168, 170, 172, 173	2006	60	Isabela, PR, USA 18°30′2.81″ N–67°01′27.66″ W	Tropical
FSP125–135, 139, 140, 142–147, 150–152, 154, 155	2010			
FSP61, 62, 66, 67, 80, 87, 91, 92, 94, 95, 102-104, 108, 109	2011			
FSP68, 70–74, 76, 78, 225	2012			
FSP195-210	2012	16	Mayaguez, PR, USA 18°12′4.84″ N–67°8′42.56″ W	Tropical
FSP213, 220	2012	2	Pioneer Seeds field, Georgia 31°42′53.68″ N–83°15′9.54″ W	Humid Subtropical
FSP244–246, 248–255	2013	11	Corozal, PR, USA 18°20′27.82″ N–66°19′0.62″ W	Tropical
FSP256–259, 261, 263	2013	6	Wharton, TX, USA 29°18′41.90″ N–96°06′9.86″ W	Humid subtropical and temperate
FSP265, 267–270, 272	2013	6	Winterville, NC, USA 35°31′44.58″ N–77°24′3.87″ W	Humid subtropical
FSP276–281	2014	16	Cairo, GA, USA 30°52′39.04″ N–84°12′7.70″ W	Humid subtropical
FSP292–301	2015			
FSP282–291	2015	10	Tifton, GA, USA 31°27′28.79″ N–83°30′21.59″ W	

^a^ Isolates of *C. sublineola* sequenced, year of collection, location, coordinates, and climatic zones of the sites. Source: www.britannica.com (accessed on 17 October 2023) Köppen Climate Classification [33].

**Table 3 jof-10-00003-t003:** Analysis of variance for the severity ratings of the thirty *Colletotrichum sublineola* isolates inoculated individually on the eighteen host differentials.

Source	DF	Sum of Squares	Mean Square	Pr. > F
Isolate ^a^	29	29.57	1.02	<0.0001 ***
Differential ^b^	17	252.12	14.83	<0.0001 ***
Isolate × differential	493	231.96	0.47	<0.0001 ***

^a^ Isolates: (Georgia: FSP213, 276, 277, 278, 279, 280, 281, 284, 289, and 299; North Carolina: FSP265 and FSP267; Puerto Rico: FSP71, 76, 92, 198, 199, 200, 201, 208, 244, 245, 248, 250, and 252; Texas: FSP36, 53, 182, and 237). ^b^ Sorghum differentials: RTx2536, SC748-5, BTx398, TAM428, RTx430, Brandes, SC112-14, Theis, BTx378, SC326-6, SC283, BTx623, SC328C, SC414-12E, PI570841, PI570726, PI569979, and IS18760. *** Highly significant at probability level of 1%.

**Table 4 jof-10-00003-t004:** Pathotype designation of 30 isolates based on their virulence pattern on the 18 sorghum differentials ^a^.

	Pathotype Designation	Sorghum Differential Lines																	
Isolate		BTx623	TAM428	PI570841	SC326-6	RTx2536	SC328C	Theis	IS18760	BTx398	SC414-12E	PI569979	PI570726	BTx378	SC283	QL3 (IND)	BRandes	SC748-5	SC112-14
200	**9**	S	S	S	S	S	S	R	S	S	S	R	R	S	S	R	R	R	R
70	**3**	S	S	S	R	R	S	S	S	S	S	R	S	R	R	S	R	R	R
248	**16**	S	S	S	S	R	S	S	R	S	R	S	R	R	S	R	R	R	R
280	**23**	S	S	S	S	S	R	R	S	S	R	R	S	S	R	R	R	R	R
277	**20**	S	S	S	S	S	R	S	S	R	R	R	S	S	R	R	R	R	R
237	**13**	S	S	S	S	S	S	S	R	R	R	S	R	R	R	S	R	R	R
244	**14**	S	S	S	S	S	S	S	R	S	R	S	R	R	R	R	R	R	R
267	**18**	S	S	S	S	S	R	R	S	S	S	R	R	R	R	R	R	R	R
276	**19**	S	S	S	S	S	R	S	S	R	R	R	R	S	R	R	R	R	R
198	**7**	S	S	S	S	S	S	R	S	R	S	R	R	R	R	R	R	R	R
199	**8**	S	S	S	R	S	S	S	S	R	R	R	R	R	R	R	R	R	R
252	**17**	S	S	S	S	R	S	R	R	S	R	S	R	R	R	R	R	R	R
213	**12**	S	S	S	S	S	R	S	R	R	R	S	R	R	R	R	R	R	R
201	**10**	S	S	S	S	S	S	R	R	R	S	R	R	R	R	R	R	R	R
279	**22**	S	S	S	S	R	R	R	R	S	R	R	R	S	R	R	R	R	R
278	**21**	S	S	S	S	S	R	R	R	R	R	R	S	R	R	R	R	R	R
182	**6**	S	S	S	R	R	S	R	R	R	S	R	R	R	R	R	R	R	R
71	**4**	S	S	S	R	R	S	S	R	R	R	R	R	R	R	R	R	R	R
284	**25**	S	S	S	R	S	R	R	R	R	R	R	R	R	S	R	R	R	R
281	**24**	S	S	S	S	R	R	R	R	R	R	R	S	R	R	R	R	R	R
299	**27**	S	S	S	S	R	R	R	R	R	R	S	R	R	R	R	R	R	R
36	**1**	S	S	S	S	R	R	R	R	R	S	R	R	R	R	R	R	R	R
289	**26**	S	R	S	S	R	R	R	R	R	R	S	R	R	R	R	R	R	R
245	**15**	S	S	S	S	R	R	R	R	R	R	R	R	R	R	R	R	R	R
250	**15**	S	S	S	S	R	R	R	R	R	R	R	R	R	R	R	R	R	R
53	**2**	S	S	R	R	R	R	S	R	R	R	R	R	R	R	R	R	R	R
76	**5**	S	S	S	R	R	R	R	R	R	R	R	R	R	R	R	R	R	R
92	**5**	S	S	S	R	R	R	R	R	R	R	R	R	R	R	R	R	R	R
265	**5**	S	S	S	R	R	R	R	R	R	R	R	R	R	R	R	R	R	R
208	**11**	S	S	R	R	R	R	R	R	R	R	R	R	R	R	R	R	R	R

^a^ FSP213, FSP276, FSP277, FSP278, FSP279, FSP280, FSP281, FSP284, FSP289, FSP299, FS265, FSP267, FSP70, FSP71, FSP76, FSP92, FSP198, FSP199, FSP200, FSP201, FSP208, FSP244, FSP245, FSP248, FSP250, FSP252, FSP36, FSP53, FSP182, and FSP237.

## Data Availability

Data are contained within the article.

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
