# Peer review of "Genetic Diversity and Classification of Colletotrichum sublineola Pathotypes Using a Standard Set of Sorghum Differentials"

_jof, 2023, doi:10.3390/jof10010003_

Round 1
Reviewer 1 Report
Comments and Suggestions for Authors
1. As shown in Table 1, there are so many sorghum differentials. Is there a set of standard sorghum differentials to distinguish pathotypes of Colletotrichum sublineola? This is of the most importance.
2. According to the results, there are so many pathotypes. Why not all the isolates were used for pathotype determination?
3. In the note of Table 2, what in Table 2 are from the source?
4. In the note of Table 3, the full names of the isolates should be provided.
5. The full names of the isolates in Table 4 should be provided.
6. In the References section, please check the Journal name.
Author Response
Reviewer 1.
We would like to extend our sincere gratitude to you for your valuable comments and suggestions. The manuscript was amended accordingly.
- As shown in Table 1, there are so many sorghum differentials. Is there a set of standard sorghum differentials to distinguish pathotypes of Colletotrichum sublineola? This is of the most
Authors’ response: Currently, there is no common set of differentials. However, this set was compiled by the authors (Prom et al. 2012) primarily for that reason to be used among researchers to obtain comparable data. The set has now been used in the USA, Africa, and Puerto Rico.
- According to the results, there are so many pathotypes. Why not all the isolates were used for pathotype determination?
Authors’ response: Due to time, labor, and availability of space limited our efforts to evaluate more isolates. However, future research will include more isolates.
- In the note of Table 2, what in Table 2 are from the
Authors’ response: Table 2 shows the sequenced isolates, year of collection, location, coordinates, and climatic zones of the sites. Footnotes included to elucidate the details of the table.
- In the note of Table 3, the full names of the isolates should be
Authors’ response: The name/identifier of each C. sublineola isolate (FSP + number) is provided in the footnotes.
- The full names of the isolates in Table 4 should be
Authors’ response: The name/identifier of each C. sublineola isolate (FSP + number) is provided in the footnotes.
- In the References section, please check the Journal
Authors’ response: All Journal names confirmed.
Reviewer 2 Report
Comments and Suggestions for Authors
The study on genetic diversity and classification of C sublineola pathotypes provides more information on this important topic in the anthracnose disease caused by this fungus. The approach to addressing the topic is interesting.To improve the reading of this work it is necessary that the tables be more illustrative. My recommendation is that they be analyzed to give a better distribution of the information collected.
Comments on the Quality of English LanguageReading the article in the English language is easy to understand, focusing on scientific expressions. There is no objection to the quality of English language skills.
Author Response
Reviewer 2.
We would like to extend our sincere gratitude to you for your valuable comments and suggestions. The manuscript was amended accordingly.
The study on genetic diversity and classification of C. sublineola pathotypes provides more information on this important topic in the anthracnose disease caused by this fungus. The approach to addressing the topic is interesting. To improve the reading of this work it is necessary that the tables be more illustrative. My recommendation is that they be analyzed to give a better distribution of the information collected.
Authors’ response: Data on anthracnose rating were analyzed. Table 3 shows the analysis of variance for the isolates, differentials, and the interactions. In addition, footnotes were provided in each table to elucidate the details.
Reviewer 3 Report
Comments and Suggestions for Authors
The study is scientifically important and should be publish. However, currently the manuscript is difficult to read.
1. The authors need to clearly explain what their aims and objectives are.
As it is currently, is is very difficult to read: To effectively deploy resistance sources, knowledge on the pathotypes of C. sublineola in a region is essential. The range of pathotype variation in C. sublineola is determined by 72 using a set of sorghum genotypes known to react differently to anthracnose (Table 1). Given that new virulent pathotypes in C. sublineola will occur, their monitoring is of paramount importance for host plant resistance in sorghum. In this study, we report the sequencing of 140 C. sublineola isolates to determine the genetic diversity through SNP markers, the selection of 30 diversified isolates collected from Georgia, North Carolina, Puerto Rico, and Texas to establish the C. sublineola pathotypes using a set of standard 18 sorghum differentials. (Lines 71 -78).
2. The authors need to clearly indicate in their materials and Results section what plants and isolates were used. For example, information around Table 1. Also, using footnotes can help explain table headings and information.
3 Please cite where necessary, e.g., Tassel 5.0. And explain clearly what was done. For example, the authors can expain in more detail "were used in this study and are described in detail by Prom et al. [18]" (Line 130).
4. In Figure 1, why only include K2 and K9. Should show K2 - K9.
5. Figure 1C can be a separate figure. Please also explain abbreviations.
6. Please explain clearly in the Discussion how the diversity and virulence are linked. And linked results with previous literature.
Comments on the Quality of English LanguageGood.
Author Response
Reviewer 3.
We would like to extend our sincere gratitude to you for your valuable comments and suggestions. The manuscript was amended accordingly.
- The authors need to clearly explain what their aims and objectives are.
Authors’ response: Agreed. Clearly stated in the last paragraph in the Introduction Section beginning with “To effectively deploy resistance………….. genetic diversity through SNP markers.”
- The authors need to clearly indicate in their materials and Results section what plants and isolates were For example, information around Table 1. Also, using footnotes can help explain table headings and information.
Authors’ response: Agreed. Footnotes included in all the tables to elucidate the details of each table. In the Materials section 2.3 and 2.4 and Results section 3.2, the sorghum differentials and isolate (FSP+number) names are provided.
- Please cite where necessary, g., Tassel 5.0. And explain clearly what was done. For example, the authors can expain in more detail ''were used in this study and are described in detail by Prom et al. [18]" (Line 130).
Authors’ response: TASSEL 5.0 is cited [31}.
- In Figure 1, why only include K2 and Should show K2 - K9.
Authors’ response: The population structure analysis based on the delta K values suggest the existence of two populations, while the analysis based on the LN probability of the data suggest the existence of nine populations (8 populations and 1 admixture group). Both analyses are correct and provide valuable information, thus, we plotted the Structure output for both K values. Also, these two graphs allow to understand how the population structure disrupt into 8 populations.
- Figure 1C can be a separate Please also explain
Authors’ response: Figure 1C is to support and understand the results found in the structure analysis (i.e. Fig 1A and 1B). Figure 1C shows which samples belong to the different populations when the diversity is divided into two or nine groups. Since admixture samples were not included, the eight circles represent the eight populations, and their color are associated with the K9 Structure graph. Certainly, having these three graphs together provides a better understanding the population structure of C. sublineola.
- Please explain clearly in the Discussion how the diversity and virulence are And linked results with previous literature.
Authors’ response: In the Discussion section, lines 327-339 beginning with “In the present study………….on some host pathosystems” cited studies that endeavor to elucidate some levels of association or not.
Round 2
Reviewer 3 Report
Comments and Suggestions for Authors
The authors need to check for minor errors. For example, "Recently, Koima et al (14) evaluated seven C. sublineola isolates with different morphological" (Lines 56-57), should read "Recently, Koima et al (14) evaluated seven C. sublineola isolates with different morphological".
The authors should try to improve the tables. Format and clarity.
Author Response
File attached
